# Immunorthodontics: Role of HIF-1α in the Regulation of (Peptidoglycan-Induced) PD-L1 Expression in Cementoblasts under Compressive Force

**DOI:** 10.3390/ijms23136977

**Published:** 2022-06-23

**Authors:** Jiawen Yong, Sabine Gröger, Joerg Meyle, Sabine Ruf

**Affiliations:** 1Department of Orthodontics, Faculty of Medicine, Justus Liebig University Giessen, D-35392 Giessen, Germany; sabine.e.groeger@dentist.med.uni-giessen.de (S.G.); sabine.ruf@dentist.med.uni-giessen.de (S.R.); 2Department of Periodontology, Faculty of Medicine, Justus Liebig University Giessen, D-35392 Giessen, Germany; joerg.meyle@dentist.med.uni-giessen.de

**Keywords:** *Porphyromonas gingivalis*, peptidoglycan, cementoblasts, HIF-1α, PD-L1, immunorthodontics

## Abstract

Patients with periodontitis undergoing orthodontic therapy may suffer from undesired dental root resorption. The purpose of this in vitro study was to investigate the molecular mechanisms resulting in PD-L1 expression of cementoblasts in response to infection with *Porphyromonas gingivalis* (*P. gingivalis*) peptidoglycan (PGN) and compressive force (CF), and its interaction with hypoxia-inducible factor (HIF)-1α molecule: The cementoblast (OCCM-30) cells were kinetically infected with various concentrations of *P. gingivalis* PGN in the presence and absence of CF. Western blotting and RT-qPCR were performed to examine the protein expression of PD-L1 and HIF-1α as well as their gene expression. Immunofluorescence was applied to visualize the localization of these proteins within cells. An HIF-1α inhibitor was added for further investigation of necroptosis by flow cytometry analysis. Releases of soluble GAS-6 were measured by ELISA. *P. gingivalis* PGN dose dependently stimulated PD-L1 upregulation in cementoblasts at protein and mRNA levels. CF combined with *P. gingivalis* PGN had synergistic effects on the induction of PD-L1. Blockade of HIF-1α inhibited the *P. gingivalis* PGN-inducible PD-L1 protein expression under compression, indicating an HIF-1α dependent regulation of PD-L1 induction. Concomitantly, an HIF-1α inhibitor decreased the GAS-6 release in the presence of CF and *P. gingivalis* PGN co-stimulation. The data suggest that PGN of *P. gingivalis* participates in PD-L1 up-regulation in cementoblasts. Additionally, the influence of compressive force on *P. gingivalis* PGN-induced PD-L1 expression occurs in HIF-1α dependently. In this regard, HIF-1α may play roles in the immune response of cementoblasts via immune-inhibitory PD-L1. Our results underline the importance of molecular mechanisms involved in bacteria-induced periodontics and root resorption.

## 1. Introduction

Compression is one hallmark of the specific microenvironments induced by orthodontic compressive force (CF) during orthodontic tooth movement (OTM), due to its capability to affect the cementoblasts homeostasis [1]. During OTM, compressive force triggers inflammatory osteoclastogenic and remodeling processes in the periodontal ligament (PDL) to further create an immune suppressive hypoxic microenvironment, which impairs the process of OTM [2]. The role of compressive force on the physiology of cementum has received increasing interest because orthodontically induced inflammatory root resorption (OIIRR) is a serious side effect of orthodontic tooth movement [3]. Cementoblasts are the responsible cell population for the regulation of cementum homeostasis [4] as well as for the prevention of OIIRR via the involvement in innate immune defense [5]. These cells thus have been intensively investigated in basic orthodontic research especially with regard to their responses to orthodontic compressive forces or periodontal pathogens and their components such as peptidoglycan.

*Porphyromonas gingivalis* (*P. gingivalis*), a Gram-negative anaerobe, has been implicated as one of keystone pathogens in the progression of chronic periodontitis [6]. Due to the increased identification of Gram-negative bacteria present in the root surface of orthodontic tooth [7], studies of the underlying mechanisms for leading to OIIRR by components of this pathogen have attracted attention. One component of the *P. gingivalis* cell wall is peptidoglycan (PGN). *P. gingivalis* PGN is a ubiquitous bacterial component that, despite its well-known powerful pro-inflammatory properties, also may play a role in immune defense responses [8]. It is known to stimulate abundant PD-L1 protein expression in human gingival keratinocytes [9], but this has not yet been demonstrated in cementoblasts. Importantly, the molecular and cellular mechanisms by which the periodontal pathogenic species, *P. gingivalis*, interact with orthodontic forces (CF) are not yet understood in cementoblasts.

In recent years, we raised for the first time a hypothetic theory that the human immune system can be modified to influence the orthodontic-treated tooth root during OIIRR. One possible mechanism is to target the programmed death-ligand 1 (PD-L1) [10]. It is also known as B7-H1 or CD274, a type-I transmembrane protein containing an N-terminal extracellular domain that acts as an immune checkpoint inhibitor to suppress the immune response through interaction with the programmed death receptor (PD)-1 that is expressed in T-cells [11], B-cells, NK cells, dendritic cells and monocytes [12]. Under physiological conditions of in vivo and in vitro models, PD-L1 can be detected in a variety of cell types, including epithelial and endothelial cells, and has been demonstrated to be overexpressed upon immune activation conditions, such as inflammation [13]. Keir et al. (2007) reported that PD-L1 plays a pivotal role in maintaining inhibitory signals to PD-1 expressing T-cells, leading to immune system impairment [14]. Thus far, PD-L1 expression on cementoblasts should be reasonably verified as a biomarker of our theory. We hypothesize that the *P. gingivalis* cell wall component, peptidoglycan, is responsible for inducible PD-L1 expression on cementoblasts. Additionally, the elevated expression of PD-L1 on cementoblasts could be strongly correlated with unfavorable hemeostasis regulation of cementoblasts. Furthermore, we aimed to demonstrate that HIF signaling participates in the regulation of PD-L1 in cementoblasts after application of CF. Therefore, immunotherapies targeting PD-L1 signaling could be a possible treatment for OIIRR because of its ability to induce durable anti-inflammatory immune responses in the group of patients during orthodontic treatment.

The hypoxia-inducible PD-L1 expression has been revealed in multiple primary and various carcinoma cell types [15,16]. The hypoxia-induced transcription factors (HIF) are crucial regulators of the transcriptional responses to orthodontic forces [17] and have been reported to regulate the apoptosis process of cementoblasts [2]. However, it is currently largely unknown whether HIF-1α regulates PD-L1 expression in cementoblasts. These observations underline the vital importance of understanding the mechanism how PD-L1 expression is regulated in cementoblasts.

## 2. Materials and Methods

### 2.1. OCCM-30 Cell Culturing

Immortalized murine cementoblast cell line, OCCM-30, was kindly provided by Prof. M. J. Somerman (University of Washington, Washington, DC, USA) [18] and Prof. J. Deschner and Dr. M. Nokhbehsaim (University of Bonn, Bonn, Germany). Cells were maintained in complete growth medium (α-MEM, Gibco, Gaithersburg, MA, USA) plus 10% fetal bovine serum (FBS) (10270-106, Gibco), 100 units/mL penicillin, and 100 μg/mL streptomycin (Gibco) in a 5% CO_2_ humidified incubator. The complete growth medium was changed three times a week. Cells were seeded at a density of 1 × 10^6^ cell/well in 6-well plates (657160, Greiner bio-one, Frickenhausen, Germany) until proliferatory outgrowth of adherently growing cementoblasts was observed. The cells used in the study was from 3rd to 5th passages.

### 2.2. Compressive Force (CF) Application

Cementoblast cultures were compressed by CF after various time periods (1, 2, 4, 6, 8, 12 and 24 h). The present in vitro CF model was performed as described previously [19,20]. In brief, OCCM-30 cells were pre-incubated for 14 h in starvation culture medium containing 0.5% FBS. Then, a glass cylinder (35 mm in diameter) with smooth surfaces was placed over the confluent cell layers for 24 h. The magnitude of compressive force applied was 2.4 gf/cm^2^. Previous investigations have proved that this mechanical compressive stress can effectively be applied by utilized in this experiment [1]. Cells cultured without glass disc served as the control cells.

### 2.3. Peptidoglycan Isolation and Stimulation

Peptidoglycan was isolated from *P. gingivalis* W83 culture at the late logarithmic state. Bacteria *P. gingivalis* were centrifuged at 8000× *g* for 10 min (Heraeus, Hanau, Germany). Afterwards, the pellet was resuspended in 10% trichloroacetic acid and incubated at 4 °C for 30 min. Cells were washed three times with 1× phosphate-buffer saline (PBS) (Gibco), then pipetted into boiling Sodium dodecyl sulfate (SDS) for three hours and stirred overnight. To pellet the peptidoglycan polymers, 160 mL of the prepared sample was centrifuged at 150,000× *g* for 1 h at room temperature (RT) in an AH-629 swinging bucket rotor in a Wxt ultracentrifuge (Thermo Fisher, Leipzig, Germany). The pellet was resuspended in 10% PBS and washed three times, then resolved in 10 mM Tris-HCl. Proteinase K (Sigma-Aldrich, Schnelldorf, Germany) at 5 ng/mL was supplied and kept overnight. Finally, samples were spun down by 150,000× *g* at RT for 1 h.

For experiments, cementoblasts were stimulated with *P. gingivalis* PGN (1, 5, 10, 20, 50, 100 and 200 μg/mL) for the indicated time periods (4, 8, 12, 24 and 48 h).

### 2.4. Chemical Inhibitor

For the HIF-1α signaling molecule inhibition experiments, 20 nM (7.7 μg/mL) HIF-1α inhibitor (IDF11774) (MedChemExpress, Monmouth Junction, NJ, USA) were used. The same amount of DimethyIsulfoxide (DMSO) (Sigma-Aldrich, Burlington, NJ, USA) was added as negative-control group. The inhibitor was pre-incubated for 2 h prior to stimulation with *P. gingivalis* PGN and CF.

### 2.5. Quantitative Real-Time Polymerase Chain Reaction (RT-qPCR)

For RNA isolation, treated-cells were first washed three times with 1× PBS (Gibco) and then extracted using commercial Nucleospin^@^ RNA plus (MACHEREY-NAGEL, Dueren, Germany). After isolation, the quality and concentrations of the eluted RNA were verified and analyzed photometrically by optical density (OD) A260/280 nm ratio readings (Nanodrop 2000, Thermo Fisher). The cDNA synthesis was performed using iScript^TM^ cDNA Synthesis Kit (Bio-Rad, Feldkirchen, Germany). RT-qPCR was performed with SsoAdvanced^TM^ Universal SYBR Green Supermix (Bio-Rad). The following primers were used: *CD274* (PD-L1, qMmuCED0044192, Bio-Rad), *HIF-1α* (qMmuCID0005501, Bio-Rad). For the normalization of target genes (Rel. mRNA), reference gene *PPIB* (qMmuCED0047854, Bio-Rad) which have been shown to be stably expressed in OCCM-30 cells were used [21]. A CFX96^TM^ Real-Time System (C1000^TM^ Thermal Cycler, Bio-Rad, Hercules, CA, USA) was used with a RT-qPCR protocol: an initial denaturation phase of 95 °C for 30 s, followed by 39 cycles of 95 °C for 15 s, and 60 °C for 30 s with a plate read every 0.5 °C increment after holding the temperature for 5 s with continuous fluorescence acquisition.

### 2.6. PCR Data Analysis

The quantitative results of the RT-qPCR were analyzed by using the comparative CT (^ΔΔCq^) method. The relative gene expression of targets (2^−ΔΔCq^) was calculated by normalizing to *PPIB* relative to non-stimulated control cells.

### 2.7. Immunofluorescence (IF)

The HIF-1α and PD-L1 expression within OCCM-30 cells was detected by IF method. Briefly, cells were fixed with 4% paraformaldehyde (pH 7.4, Sigma-Aldrich) and permeabilized with 0.5% Triton™ X-100 (Thermo Fisher) at RT. Then, cells were incubated in blocking buffer (Cell Signaling Technology, Leiden, The Netherlands) for 30 min at RT and washed with 1× PBS supplement with 0.02% Tween-20 (Sigma-Aldrich) (PBST) for every step. Then, cells were incubated with PD-L1 polyclonal antibody (Invitrogen, Leipzig, Germany) (dilution 1:250) or HIF-1α polyclonal antibody (Invitrogen) (dilution 1:400) at 4 °C overnight. After washing three times with 1× PBST, the samples were incubated with DyLight^@^ 488 goat anti-rabbit polyclonal antibody (Abcam, Cambridge, UK) (dilution 1:500) that can be conjugated with fluorescein isothiocyanate, for 1 h in the dark atmosphere. Nuclear deoxyribonucleic acid was stained was performed using a fluorescent Mounting Medium containing 4′,6-diamidino-2-phenylindole (DAPI) (Abcam, Berlin, Germany). Slides were analyzed using a high-resolution fluorescence microscope (Leica Inc., Wetzlar, Germany) and photographed.

### 2.8. Annexin-V FITC/Propidium Iodide (PI) Staining

Both adherent and floating cells were collected and transferred into 1.5 mL reaction tubes. The samples were washed twice with FACS buffer (BD Pharmingen, Heidelbery, Germany), centrifuged at 500× *g* and the supernatant were aspirated. Samples were then resuspended at a concentration of 1 × 10^6^ cells/mL in FACS buffer (BD Pharmingen, Germany) and transferred (100 μL) into round-bottom 12 × 75 mm polystyrene Falcon tubes (Thermo Fisher). For necroptosis detection, cells were stained by mixing 5 μL of Annexin-V FITC and 5 μL propidium iodide (PI) in staining buffer. Following incubation at RT and protected from light for 15 min, 400 μL of staining buffer was finally added to every tube. Samples were analyzed using a FACS Vantage Flow Cytometer (Sony Biotechnology, Berlin, Germany).

### 2.9. Quantification of Necroptotic Cell Populations

The data was obtained using the SP6800 Spectral Analyzer software version 2.0.2 (Sony Corporation, Tykyo, Japan) for analysis. Data were analyzed using the FlowJo software version 10 (Treestar, Ashland, OR, USA). To summarize, the 488 nm laser was used for excitation. Debris and doublets were barred from entering. The channels of bright field (430–480 nm), Annexin-V FITC (505–560 nm) and PI (592–642 nm) were measured and at least 1 × 10^4^ single cell events were recorded per sample. Because the emission spectra of FITC and PI overlap, color compensation was required. Additional single-labeled samples were prepared, single stained with FITC or PI, respectively.

The gating strategy was determined by the fluorescence intensity of Annexin-V FITC and PI. These cells are classified as double negative (healthy), Annexin-V FITC positive (apoptotic) and double positive (necroptotic) cells.

### 2.10. Western Blotting (WB)

The cultured OCCM-30 cells were washed three times with 1× PBS, followed by lysis using Pierce^TM^ RIPA buffer (Thermo Fisher) supplemented with 3% Protease and Phosphatase Inhibitor Cocktails (Thermo Fisher). Protein concentration was measured using the bicinchoninic acid assay (Pierce^TM^ BCA, Thermo Fisher) in a Nanodrop 2000 Spectrophotometer. Then, 20 μg of protein lysates were loaded on the gels for electrophoresis on 4–20% Precast Gels (Mini-PROTEAN^@^ TGX^TM^, Bio-Rad) and blotted onto nitrocellulose membranes (Bio-Rad) using a turbo blotting semi-dry device (Trans-Blot^@^ Turbo^TM^ Transfer System, Bio-Rad).

The blotted membranes were incubated with 5% non-fat milk blocking buffer (ROTH, Germany) for 1 h at RT and then incubated with the primary antibodies: PD-L1 polyclonal antibody (Invitrogen) (dilution 1:1000) or HIF-1α polyclonal antibody (Invitrogen) (dilution 1:1000). Β-actin (Abcam, Germany) (dilution 1:2000) was used to standardize loading. Ponceau S staining (Sigma-Aldrich) was applied to profile the bands of whole protein. The secondary antibody Polyclonal Goat Anti-Rabbit IgG HRP (Dako, Santa Clara, CA, USA) was used at a dilution of 1:2000. All blots were incubated with Amersham ECL WB Detection Reagents (Cytiva, Tokyo, Japan), followed by detection using Amersham Hyperfilm (Cytiva) on X-Ray Film Processor (PROTEC GmbH, Oberstenfeld, Germany). Finally, the X-rays were scanned and analyzed by the ImageJ program (SciJava, NIH, Washington, DC, USA).

### 2.11. GAS-6 Production (ELISA)

For the quantification of the GAS-6 release in the culture supernatants, a highly sensitive mouse GAS-6 ELISA kit (BIOZOL, Eching, Germany) was used following the manufacturer’s instructions. Cell supernatants collected from 3 independent experiments with sum of 6 biological replicates were performed. The relative OD_405_ was recorded for every sample by calculating the mean OD_405_ replicate of each standard GAS-6 serial dilution vs. the respective standard GAS-6 concentrations. The final concentration of soluble GAS-6 in cell supernatant was interpolated by utilizing the linear regression of the relative OD_405_ against the calibration curve.

### 2.12. Statistical Analysis

GraphPad Prism 8.0 software was used for statistical analysis (GraphPad software Inc., San Diego, CA, USA). To determine statistically significant differences between two groups, all values are expressed as means ± standard deviation (SD) and analyzed using student’s *t*-test for unpaired samples. For multiple comparisons, one-way ANOVA with post turkey evaluation was used. At a *p* value of < 0.05, differences were considered statistically significant. Each experiment was carried out three times.

## 3. Results

### 3.1. P. gingivalis PGN Triggers PD-L1 Up-Regulation in Cementoblasts

The kinetics of PD-L1 expression in cementoblasts after infection with various concentrations of *P. gingivalis* PGN were verified by all used methods. We found that stimulation of cementoblasts with *P. gingivalis* PGN led to the upregulation of PD-L1 protein in a concentration-dependent manner, showing a peak of PD-L1 up-regulation at 100 μg/mL as revealed by WB (Figure 1A–E). The IF staining of PD-L1 showed dramatically increased expression in *P. gingivalis* PGN (100 μg/mL)-infected cells compared to non-stimulated cells (Figure 1F). The gene expression analysis by RT-qPCR revealed that the mRNA level of *PD-L1* was significantly higher after 24 h in response to 100 μg/mL *P. gingivalis* PGN stimulation compared to the start of the incubation (time point 0) (Figure 1G).

Overall, these data clearly indicated that *P. gingivalis* PGN significantly up-regulated PD-L1 molecule expression in cementoblasts in a dose- and time-dependent manner, beginning at 4 h, showing a peak after 8 h that then attenuates after 12 h up to 48 h.

### 3.2. Compressive Force (CF) Triggers PD-L1 and HIF-1α Expression in OCCM-30 Cells

As compression is one of the major components of orthodontic-induced microenvironment, the effect of CF on the expression of immune checkpoint receptor ligand (PD-L1) was for the first time investigated on cementoblasts. The results show that an application of 2.4 gf/cm^2^ of CF significantly increased the PD-L1 expression at mRNA and protein levels during 8, 12 and 24 h (Figure 2B,C). Similarly, it was strongly up-regulated in CF group as observed visually through the immunofluorescence staining of PD-L1 (Figure 2D).

We further investigated whether CF could induce HIF-1α expression in OCCM-30 cells. As shown in Figure 1E, CF significantly increased the HIF-1α protein and gene expression in cementoblasts (Figure 1E,F). We also observed that HIF-1α was abundant in compressed cells (Figure 1G). Therefore, our results show that compressive force resulted in slight PD-L1 up-regulation and promoted HIF-1α expression in cementoblasts.

### 3.3. Compression Mediates P. gingivalis PGN-Triggered PD-L1 Expression on Cementoblasts

WB analysis showed that PD-L1 was up-regulated in *P. gingivalis* PGN-infected-cementoblast cells (Figure 3A–D). Importantly, especially after 24 h co-stimulation with CF and *P. gingivalis* PGN, the PD-L1 protein expression was enhanced. *P. gingivalis* PGN or CF also increased the ratio of necroptosis (11.22 ± 0.75%, 14.79 ± 1.84%, respectively) as compared to the control group (7.19 ± 0.60%) (Figure 3E,F). *P. gingivalis* PGN combined with CF significantly increased necroptosis of cementoblasts (39.55 ± 4.51%) (Figure 3E,F). Compressive force was shown to regulate the *P. gingivalis* PGN-induced PD-L1 expression and to participate in the necroptosis of cementoblasts.

### 3.4. Correlation of P. gingivalis PGN-Induced PD-L1 and HIF-1α in Cementoblasts under Compressive Force

To explore whether the PD-L1 expression induced by *P. gingivalis* PGN in cementoblasts depends on HIF-1α, the HIF-1α inhibitor IDF11774 was used. The WB results showed that blockade of HIF-1α under compressive force significantly abrogated the *P. gingivalis* PGN-induced PD-L1 protein up-regulation (Figure 4A,B). IF staining showed a significant inhibition impact of HIF-1α inhibition on inducible PD-L1 expression in the presence of *P. gingivalis* PGN stimulation (Figure 4C). Moreover, after 24 h co-stimulation with IDF11774, it was observed that the necroptosis of cells was induced (Figure 4C,D). In the presence of compressive force, IDF11774 increased the percentage of necroptotic cells. The percentage of necroptotic cells in *P. gingivalis* PGN plus CF group was also increased (Figure 4E). Thus, the evidence suggests that HIF-1α is a major regulator of *P. gingivalis* PGN-induced PD-L1 mRNA and protein expression under compressive force.

### 3.5. HIF-1α Is Involved in the Immune Modulation of Cementoblasts via Reduction in GAS-6 Secretion

It was described that soluble GAS-6 exerts immunomodulatory effects in the homeostasis of PDL cells. Thus, we aimed to analyze whether OCCM-30 cells can release GAS-6 and its expression can be altered by co-stimulation with *P. gingivalis* PGN and/or CF in the presence or absence of the HIF-1α inhibitor. The results shown that OCCM-30 cells have the property to release GAS-6 (27.67 ± 11.15 pg/mL), and this release is non significantly reduced usage of IDF11774 (18.04 ± 8.43 pg/mL) (Figure 4F). Treatment with *P. gingivalis* PGN did not change the GAS-6 production. Interestingly, under CF as well as CF plus *P. gingivalis* PGN conditions, IDF11774 caused a decreased release of GAS-6 to 10.72 ± 1.82 pg/mL and 19.33 ± 8.84 pg/mL, respectively. Thus, GAS-6, an immune modulator of cementoblasts, could be released and regulated through HIF-1α.

## 4. Discussion

This is the first cellular study to demonstrate that *P. gingivalis* PGN-induced inflammatory microenvironment is associated with immune checkpoint PD-L1 expression, and it revealed its interactions with compressive force. Although the involvement of PD-L1 in the morbid progression of periodontitis is strongly evidence-based, the functional contribution of PD-L1 expression in the field of orthodontics has not been investigated before. In the present study, the results indicate that *P. gingivalis* PGN significantly induced high levels of PD-L1 up-regulation on cementoblasts. Meanwhile, compressive force triggers the PD-L1 as well as HIF-1α expression. Furthermore, the PD-L1 is found to interact with HIF-1α and acts as target in response to compression on cementoblasts. Since compression is one of the major components of the orthodontic-induced microenvironment, the HIF-1α/PD-L1 pathway is hypothesized to be the major immunorthodontic modulator during OTM as well as OIIRR.

Previous studies showed that *P. gingivalis* could induce PD-L1 up-regulation in primary, as well as immortalized, human gingival keratinocytes [22]. PD-L1 expression has been found to be increased by *P. gingivalis* infection in oral squamous carcinoma cells [22], facilitating them to evade immune elimination via the interaction of PD-L1 on the surface of carcinoma cells with PD-1 on T-cells [23]. Until now, besides in different human carcinomas, PD-L1 has also been reported to be expressed on OIIRR-associated periodontal tissue cell types including human osteoblasts [24], which has a similar phenotype as cementoblasts, human osteoclasts [25], human periodontal ligament cells [26], human gingival fibroblasts [27] and human gingival keratinocytes [22]. In our study, for the first time it could be demonstrated that *P. gingivalis* PGN upregulated PD-L1 expression in cementoblasts. Application of compressive force modified this effect by enhancing the PD-L1 expression. Furthermore, it was demonstrated that HIF-1α plays an important role in the underlying mechanisms. Immune checkpoint inhibitor PD-L1 targeting PD-1 has been reported to significantly improve the survival outcome of melanoma [28]. This led to the hypothesis that during OTM the up-regulated PD-L1 participates in immune evasion. Binding of the PD-L1 ligand to its receptor PD-1 on T-cells supports the development of an immunosuppress microenvironment, resulting in T-cell exhaustion and apoptosis [29] and may thus accelerate the process of OIIRR.

Interestingly, the kinetic analysis of compressive force-induced PD-L1 has a similar trend to that expression observed upon the HIF-1α expression. Recent studies by Kirschneck et al. (2020) reported that compressive force can induce HIF-1α stabilization in periodontal ligament fibroblasts [17] and may therefore play immunomodulatory roles during compressive orthodontic strain condition. Noman et al. (2014) reported that the immunomodulatory effects of PD-L1 are correlated with HIF-1α stabilization [30]. Our present data revealed that compressive force at the magnitude of 2.4 gf/cm^2^ triggered the HIF-1α expression. However, the mRNA levels decreased rapidly from 12 h stimulation possibly due to the initiation of apoptotic processes [2]. In this study, we clarified the expression of *P. gingivalis* PGN-induced PD-L1 and correlation of PD-L1/HIF-1α in cementoblasts. Our findings indicate that *P. gingivalis* PGN and compressive force induces the PD-L1 up-regulation in cementoblast. The latter effect is likely to occur via HIF-1α engagement since the HIF-1α inhibition suppressed PD-L1 expression, triggering a decreased secretion of the immunomodulator GAS-6 (Figure 1).

Apoptotic processes are up-regulated after compression of cementoblasts [31]. Recently, significant progress has been made in the studies of non-apoptotic forms of cell death, particularly necroptosis [32]. To verify the necroptosis data, we conducted flow cytometry method to determine the percentage of necroptotic cells inside the compression- and/or *P. gingivalis* PGN-treated OCCC-30 cells population. The results demonstrated that the rate of necroptotic cells was particularly higher after 24 h of compressive force and *P. gingivalis* PGN co-stimulation. Our present data suggest that increased rates of non-physiological death of cementoblasts via necroptosis on the compressive side of tooth root may be one reason why the repair processes for OIIRR occasionally fail to initial. We also demonstrate increased necroptosis of cementoblasts in response to *P. gingivalis* PGN and CF, and the simultaneous *P. gingivalis* PGN-induced necroptosis induction is dependent on HIF-1α. Our observation may provide evidence for the downregulation of signaling pathways leading to deficient repair activity of cementoblasts. Our data also provide evidence that HIF-1α is a key modulator for the compression induced necroptosis of cementoblasts (Figure 1).

In our previous study, GAS-6 was found to contribute to PDL cells’ homeostasis by inducing *Collagen*-1, -3, *TGF*-*β*1 and *Periostin* gene expression [19]. In line with our previous data, blockade of HIF-1α suppresses the GAS-6 secretion, indicating that compressive force triggered HIF-1α/PD-L1 signalings to regulate the immune response of cementoblasts partially through soluble GAS-6 secretion.

OIIRR is an unavoidable side effect of OTM with a wide range of clinical consequences depending on its severity. It is widely acknowledged that the cementum layer covering the tooth root surface is critical in preventing OIIRR during OTM [3]. Cementoblasts lining the root surface typically repair the damaged areas in part [33]. Cementoblasts are highly differentiated PDL mesenchymal cells that form cementum [34]. However, the reason cementoblasts lose their repair function in such cases is yet unknown. With regard to the relation between OIIRR and HIF-1α/PD-L1, our data indicated that HIF-1α is important for the induction of PD-L1 in response to orthodontic forces. Knowledge acquired in the present study possibly suggests that *P. gingivalis* PGN stimulates PD-L1 upregulation and may induce tolerogenic signaling to T-cells. Certainly, CF enabled *P. gingivalis* PGN-induced stable PD-L1 expression in cementoblasts that, in turn, may facilitate antigenic escape from immune surveillance. These effects enable the assumption that this ligand is a potential anti-OIIRR target [15]. This study represents a significant step toward unraveling the recognition mechanisms for these potentially pathogenic components in the root surface during OTM. However, one of the limitations is the immortalized murine cell line used in the present study. Thus, future studies should be performed with primary human cementoblasts as well as in the in vitro rat model to verify the mechanisms. It should be kept in mind that murine cementoblasts cannot be fully comparable to human primary cementoblasts.

## 5. Conclusions

Based on the in vitro murine cementoblast model, our study suggests that *P. gingivalis* PGN up-regulated PD-L1 on cementoblasts and that this effect was maintained via HIF-1α in response to compressive force. HIF-1α activation modulates the necroptosis of cementoblasts. Blockade of HIF-1α abrogated the immune response of cementoblasts by decreasing GAS-6 release.

Our data show for the first time that immune checkpoint PD-L1 is associated with HIF-1α in cementoblasts’ immune suppression under orthodontic-induced microenvironment. Furthermore, an immunotherapy therapy for OIIRR in periodontitis individuals during OTM, by using HIF-1α along with PD-L1-targeted inhibition, may contribute to the immune response.

## Data Availability

The datasets used and/or analyzed during the current study are available and will be provided by the corresponding author on reasonable request.

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
