# Peer review of "Immunorthodontics: Role of HIF-1α in the Regulation of (Peptidoglycan-Induced) PD-L1 Expression in Cementoblasts under Compressive Force"

_ijms, 2022, doi:10.3390/ijms23136977_

Round 1

Reviewer 1 Report

The evaluation of the molecular mechanisms of   bacteria-induced periodontics is on request for modern  orthodontics.  The goal of this MS is sound as related.

Comments and Suggestions for Authors

 M&M

 OCCM-30 cells were pre-incubated for 14 h in culture medium containing 0.5% FBS before CF application? The serum deprivation is a serious stress for cells. It is well known, that serum deprivation induces a lot of signaling pathways. Did the authors control the effects of serum deprivation on parameters estimated?

Did the authors examine the effects of PGN and CF on the percent of necrotic cells (AnnV-/PI+)?

Results

As followed from the paper, the authors supposed that CF could induce hypoxia-associated events  in the cells. Have  they own or literature data on the alteration of other then HIF hypoxia-dependent molecules?

CF  affects  mechanoreception and mechanotransduction in cells. How the tensegrity root could be taken into account for the interpretation of the results presented in MS?

Discussion

The final statement in  Discussion “However, one of the limitations is the immortalized murine cell line used in the present study. Thus, future studies should be performed with primary human cementoblasts. It should be kept in mind that murine cementoblasts are not fully comparable to  human primary cementoblasts ” depreciates significantly the data presented. If the authors understood apriory the limitations of their design, why did they still continue? What is the scientific significance of such experiments?

Reviewer 2 Report

Dear authors,

This is well conducted study and a nicely written paper.

However, the hypothesis in can be more elaborated and specifically stated in the introduction of the paper.

As this is an in vitro study, future study in animal models rather than just human cementoblast model can produce help produce more conclusive results.

The conclusion should clearly state that the findings of this study is from an in vitro  murine cementoblast model. This should bring the correct perspective to the interested readers.

Reviewer 3 Report

In summary

This manuscript by Yong  et al. assessed the role of compressive force in PD-L1 expression in cementoblasts in vitro. They tried to connect compressive force with HIF-1-PD-L1 axis. The topic is interesting, but the current data did not support the conclusion well, and some parts need to be modified.

Major

1.      in Figure 1F, it looks like there is no PD-L1 expressed by cells, compared to PGN-treated cells. But RNA and protein level (Figure 1G and D), just about 1.3 times change. Which one should be right? What is the concentration you used in Figure 1F and G?

2.      Line 224, I do not think it is "dose- and time-dependent manner", since after 20ug/ml, 100ug/ml or 48 h, the concentration was decreased. you should carefully describe your findings.

3.      Figure 2B, the bands of PD-L1 were too blur, it is difficult to quantify them, why not expose them longer to get more reliable pictures?

4.      Figure 2D, There is the similar issue with Figure 1F

5.      Figure 3A-D, you should compare the differences between PGN and CF+PGN, right? What is the "n="?

6.      Figure 4A, the bands of PD-L1 were too weak. I did not see any correlation between CF and HIF-1a,

7.      Based on the above questions, you should correct your scheme 1.

Minor:

1.     You should mention where is Figure 1A in your text.

2.     Figure 2E, F, protein levels were not consistent with RNA levels, you should discuss/explain this differences.

Round 2

Reviewer 1 Report

I thank the authors for the work done.

Reviewer 3 Report

The authors have addressed all of my comments.